# Uncertainty Problems in Image Change Detection

**Wenyu Wang [1,\*], Mryka Hall-Beyer [2], Changshan Wu [1,3], Weihua Fang [4] and Walter Nsengiyumva [5]**

[1]  School of Geomatics and Urban Spatial Information, Beijing University of Civil Engineering and Architecture, Beijing 100044, China; cswu@uwm.edu
[2]  Department of Geography, University of Calgary, Calgary, AB T2N 1N4, Canada; mhallbey@ucalgary.ca
[3]  Department of Geography, University of Wisconsin-Milwaukee, Milwaukee, WI 53211, USA
[4]  Key Laboratory of Environmental Change and Natural Disaster, Ministry of Education, Academy of Disaster Reduction and Emergency Management; Faculty of Geographical Science, Beijing Normal University, Beijing 100875, China; weihua.fang@bnu.edu.cn
[5]  College of Mechanical Engineering and Automation, Fuzhou University, Fuzhou 350108, China; Nsewalt@fzu.edu.cn
\*  Correspondence: wangwenyu@bucea.edu.cn

**Abstract:** Image Change Detection (ICD) methods are widely adopted to update large area land use/cover products. Uncertainty problems, however, are well known in such techniques, and a transparent assessment is necessary. In this study, a framework was proposed for evaluating binary land change utilizing remote sensing images. First, two widely adopted ICD methods were used to establish change maps. Second, binary decisions on Change (C) and Non-Change (NC) classes were reached through thresholding on change maps. Then, results were evaluated using two sampling designs: random sampling and stratified sampling. Analysis of results suggests that (1) for random sampling, with an increasing threshold on change variables, the overall accuracy increases and shows a large variance, which is highly correlated with the C omission error; and (2) comparatively, for stratified sampling, in which two strata (i.e., C and NC) were set, the overall accuracy shows a smaller variance and is highly associated with the NC commission error. The significant trends in accuracy assessments indicate the trade-offs between the C and NC classification errors in a binary decision and can present superficial or perfunctory accuracy evaluation in certain circumstances that the causes of error sources and uncertainty problems in ICD are not fully understood.

**Keywords:** land change; evaluation; accuracy analysis; image change detection

## 1. Introduction

Land Use/Cover Change (LUCC), also known as Land Change (LC), reflects a close interaction between the economic development and environmental biodiversity [1] and shapes our decisions on sustainability. Benefitting from the availability of remote sensing image archives, many regions have established LUCC databases [2–4]. To closely monitor LUCC, researchers accordingly have sped up the pace in renewing LUCC products [4–7]. For example, a new generation of National Land Cover Database (NLCD) products named NLCD 2016 contains LUCC products for nearly two decades [8]. All these products utilize Landsat images as their main data sources [5,8]. It was reported that the NLCD 2006 had the first land change product, employing image change detection (ICD) technology [2], followed by the NLCD 2011, adopting improved solutions [5]. Similarly, the Chinese national LUCC database has products for six periods (1980s, 1995, 2000, 2005, 2008, and 2010) [3]. To utilize LUCC products in different applications, it is necessary to understand the workflow and qualities of these products.

Techniques for Image Change Detection (ICD), that automatically correlate and compare sets of images taken from the same area at different times [9], are widely adopted to automate or semi-automate the updating of LUCC products. Over the past decades, numerous ICD methods, using geospatial data, have been developed [9–16]. In this study, we only focused on bi-temporal (before–after) and binary (Change (C)/Non-Change (NC)) land change detection. To facilitate our work, we divided ICD techniques into two categories: bi-temporal image analysis and image-and-map analysis, based on whether thematic land classes are used to facilitate the image analysis or not. Bi-temporal image analysis is favorable for its high efficiency and effectiveness, by implementing analysis at a pixel level without ground truth data [11]. Nevertheless, Canty [17] pointed out that if change detection is carried out at the pixel level (as opposed to using segments or objects), error (typically >5%) may corrupt or even dominate the true change signal depending on its strength. Instead of adopting bi-temporal images, "image-and-map analysis" takes the thematic maps as auxiliary data in ICD. As images are not directly compared to each other during the land change detection process, radiometric and phenology differences could be well resolved [18]. The "Cross-Correlation" algorithm, designed by Koeln and Bissonnette [19], can be used to evaluate the differences between an existing land cover map (Date 1) and a recent single-date multispectral image (Date 2) [11,16,20]. Unlike bi-temporal image analysis, pixel-based spectral analysis can contribute to spatial analysis at an object-level by segmenting heterogeneous land surface into homogenous classes first. In image-and-map analysis, ICD can be regarded as a posterior step after image classification, so the final accuracy depends on the quality of the classified image of each date [12]. Consistent with the aforesaid, it is clear that there is no single method that is accurate enough to handle all the challenges when it comes to ICD. Therefore, researchers can combine several methods to produce LUCC products. For example, the Comprehensive Change Detection Method (CCDM), a knowledge- and trajectory-based solution, was utilized to produce the 2011 NLCD. In this technical framework, various image indices had been designed and used to either compute the change magnitude of land cover or indicate the change direction of biomass. At the same time, the thematic maps in CCDM facilitate the change decisions. In this paper, we adopted ICD techniques to automate land change detection.

Uncertainty problems are inherent parts of computer modeling [21–23]. Studies [24–28] have identified common error sources in most of the LUCC studies produced thus far. The above was corroborated with a number of other groups of researchers [29,30] who further added that most of the challenges exist without well-established solutions for this task. Among these challenges in ICD, some significant ones need to be highlighted. The primary challenge exists in the simple dichotic form of land change decision. For binary land change detection, image pixels are divided into two categories, Change (C) and No Change (NC), adopting various statistical techniques such as thresholding or hypothesis testing. Such techniques are criticized for failing to consider the change features based on type, severity, and continuity [31]. Furthermore, to select a threshold is complicated as the proper value is scene-specific and experientially selecting a value is not robust [32,33]. In many cases, values were adopted arbitrarily. For example, researchers have begun their analysis by proposing a distribution of their observations and then set their thresholds on the point of one standard deviation away from expected values (mean). Another significant challenge is that "change is a rare feature over a given landscape" [30]. According to Jensen [11], in most regional projects, the amount of actual change over a 1–5 year period is probably not greater than 10% of the total area. Such land features add to our difficulties in ICD modeling and spatial sampling. Moreover, land changes are dynamic processes. Samples need to be collected in 3 dimensional space [34]. Not only the spatial heterogeneity, but also the temporal characteristic of land change scenarios should be considered for spatial sampling, as the uncertainty of the final estimation accumulates in the trinity of spatial sampling and statistical inference [34]. As the aforementioned violations of both ICD modeling and sampling can introduce uncertainties in the final evaluation, one needs to meet many criteria to avoid the pitfalls during the assessment process for land change detection [30]. In this study, to explore uncertainty problems in ICD, two categories of ICD methods (i.e., "bi-temporal image analysis" plus "image-and-map

analysis") were adopted and binary change decisions were achieved by simulating various values of thresholds. Such decisions were evaluated based on two common sampling strategies, through which the uncertainty problems in ICD were illustrated.

## 2. Materials and Methods

### 2.1. Study Area and Data

The city of Beijing, China, is located in the northern hemisphere with latitude from 39°26′ to 41°03′ and longitude from 115°25′ to 117°30′. The area of Beijing is about 16,800 km$^2$, which is mainly composed of mountains (10,418 km$^2$) and plains (6390 km$^2$). The population of Beijing is approximately 21,516,000 [35]. As is the case for many metropolises in the world, Beijing has been facing fast urbanization trends. To explore the inter-annual land change patterns of this metropolis during the period 2010–2015, we downloaded Landsat images from the United States Geological Survey Website [36]. We considered several factors while selecting images, including (1) sensor selection; (2) anniversary date synchronization [37]; (3) band selection; (4) cloud cover < 20%; and (5) quality ≥ 7. Two Landsat satellites, namely, Landsat 5 and Landsat 7, were simultaneously in service in 2010. When Landsat 5 became obsolete in 2013, we decided to switch to Landsat 7 and Landsat 8. Unfortunately, because some images from Landsat 7 needed significant corrections due to the Scan Line Corrector (SLC) failure, we chose Landsat 8 with the Operational Land Imager (OLI). As a result, the bi-temporal images (2010-12-14 TM and 2014-12-25 OLI image) were selected for this pilot study. The spatial resolutions for these bands for the two sensors are both 30 meters. Besides Landsat images, a 2010 classification map was acquired from the GlobaLand30, "30 m Global Land Cover (GLC) data product" [38], which is a global dataset with level 1 land cover classification scheme based on the year 2010. The test area, a sub-area of 518 × 482 pixels from a Landsat image (Figure 1) in the eastern plain of Beijing undergoes rapid urban expansion [39] and was hence selected for a pilot study. In the test area, there are three land cover types: artificial surface, cultivated land, and water, respectively composing 49.6%, 49.6%, and 0.8% of the study area (Figure 2).

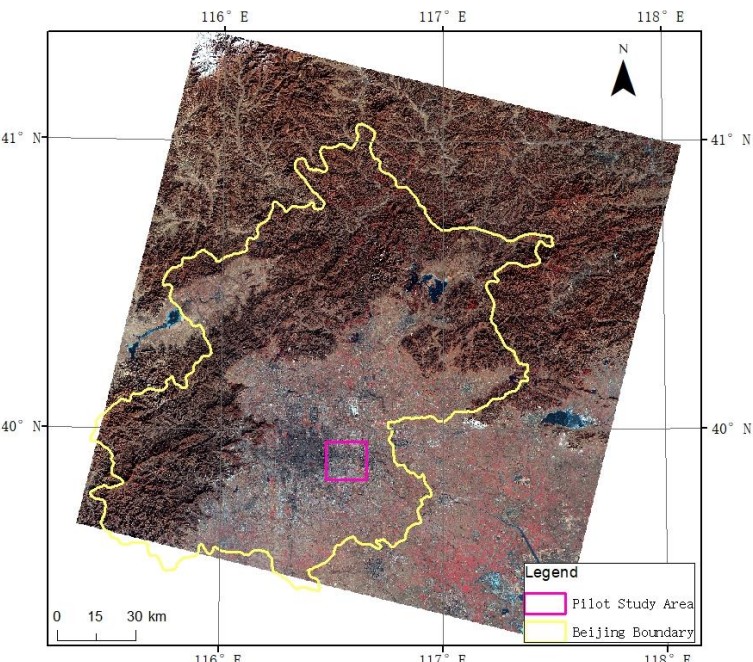

**Figure 1.** Beijing 2014-12-25 Landsat-8 image (composited using false color) and pilot study area.

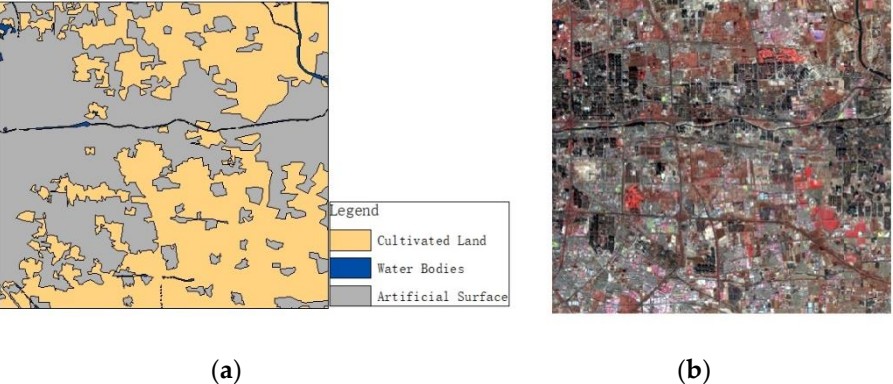

(**a**) (**b**)

**Figure 2.** Classification map and image for the study area: (**a**): 30 m Global Land Cover map; (**b**) 2010-12-14 Landsat-5 image (composited using false color).

*2.2. Methodology*

In this study, a framework was designed to explore the uncertainty problems in ICD. It was composed of three main modules (Figure 3): (1) module 1: Input; (2) module 2: Change Detection; (3) module 3: Evaluation. In module 1, geospatial data were prepared. In module 2, two ICD methods (i.e., "bi-temporal image analysis" plus "image-and-map analysis") were adopted to construct change maps, followed by a simulation on the values of thresholds to have different land change decisions. In module 3, accuracy assessments were implemented by first designing spatial sampling and computing error matrices.

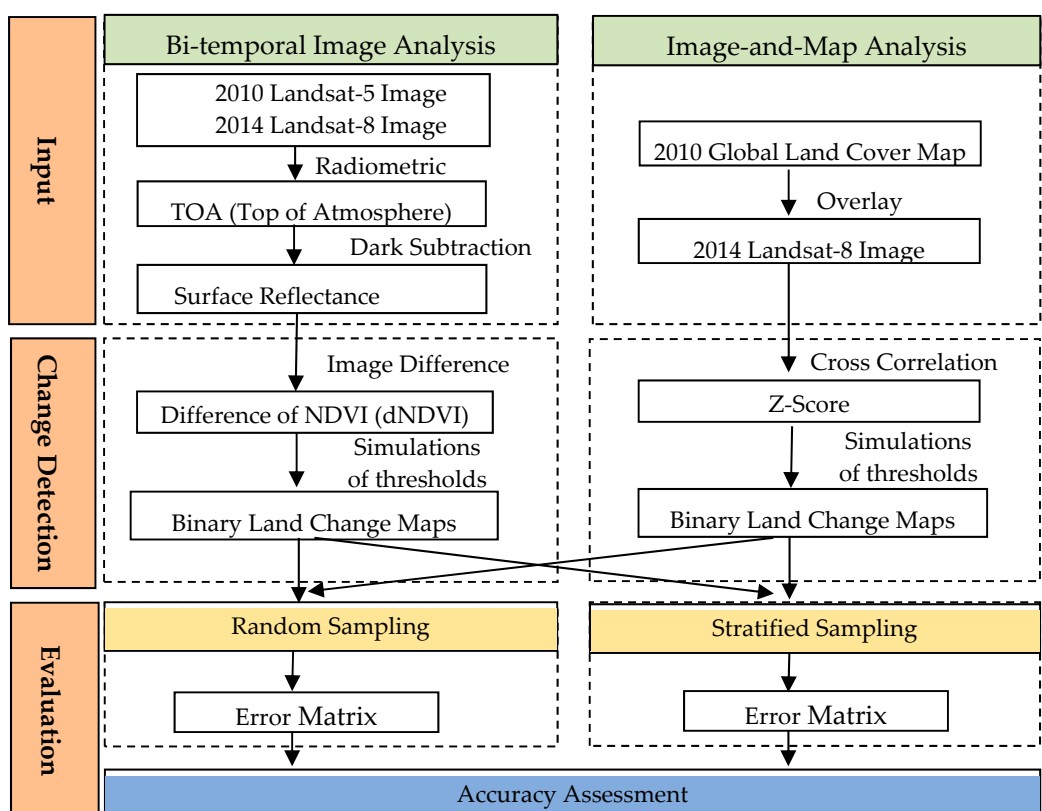

**Figure 3.** Framework of image-based land change detection and accuracy assessment (NDVI: the Normalized Difference Vegetation Index).

### 2.2.1. Change Variables

Two change variables were established adopting "bi-temporal image analysis" and "image-and-map analysis", respectively. Two pre-processing steps were taken to change the two raw images to "clear" images by removing scattering and absorption effects. Firstly, the digital numbers (DNs) of the two Landsat scenes were rescaled to the top of atmosphere (TOA) reflectance. Secondly, the surface (ground) reflectance free from the atmospheric effect was computed. Furthermore, spatial registration was checked visually. The original spatial accuracy of two original Landsat images was about one pixel within the test area. Then, the Normalized Difference Vegetation Index (NDVI), a standardized index of greenness (relative biomass), was chosen as the change variable for bi-temporal images. To be rendered into a color map easily, the NDVI values were scaled to a range of 0–200. By subtracting two NDVI images, one from the other (after image minus before image), the differencing NDVI (dNDVI) could be achieved.

Another change variable, the Z-score, was calculated according to the Cross-Correlation (CC) algorithm. First, the mean and standard deviation of the reflectance of each band on the Date 2 image (2014 OLI image) were calculated for each land class in Date 1 map (GlobaLand30). Then, the Z-score was calculated as the normalized distance from each pixel value to the "expected" value (mean) of that class (Equation (1)):

$$z = \sum_i ((B_i - \mu_{ic})/\sigma_{ic})^2 \qquad (1)$$

where $z$ is the Z-score, *i* is the band number, *c* is the land cover type, $B_i$ is the ith band of Landsat image, and $\mu_{ic}$ and $\sigma_{ic}$ are the mean and standard deviation of the ith band of the image over the c land cover type area, respectively [19].

Once change maps are constructed, land change decisions can be achieved by adopting thresholding techniques based on the assumptions made from the change magnitudes: the higher the change magnitude is, the higher the chance of having change will be.

### 2.2.2. Sampling and Accuracy Assessments

To evaluate the change detection, two common sample designs (random and stratified sampling) were adopted by selecting 100 samples, respectively. The referenced data were gathered through Google Earth, and the available high-resolution scenes in the Beijing area (2010-11-11, 2010-11-19, 2014-11-22, and 2014-11-24) were chosen. Moreover, the decision on C/NC mainly was based on the usage of lands, for example, from barren soil to factories or from buildings to the desolated area. Samples were gathered by checking the approximate extent of 5 × 5 matrix areas on the Google Earth images, taking the discrepancy of the spatial resolutions between image sources into consideration. In random sampling, 12 samples were referenced as C and the left 88 were as NC. The disproportion of C/NC samples was due to the fact that most of the area remains unchanged and only a small amount of area had changed. In stratified sampling, two strata (i.e., C and NC) were set and 50 samples were randomly drawn from each stratum. Such a strategy was adopted in an earlier study by Jin et al. [5] to produce the NLCD products. Based on the above two sample designs, an accuracy assessment was carried out by calculating the binary change/no change confusion matrix [40]. Three types of accuracy could be calculated: overall accuracy, producer's accuracy, and user's accuracy.

## 3. Results

Both the spatial and frequency distributions of the two change detectors (i.e., dNDVI and Z-score/CC) are illustrated in Figure 4. Looking at the spatial distribution of dNDVI (Figure 4a), the red color denotes high dNDVI values, and the green color denotes lower dNDVI values. Based on our personal observation, some red color areas could be visually identified as the vegetation. Figure 4b shows the histogram of dNDVI. It was a two-sided bell-shaped distribution, not only denoting the change magnitude but also showing the change directions (positive and negative values of change

values). A series of thresholds were derived using 12 decrement numbers, and thresholds were calculated by setting different deviations from the mean of dNDVI (Equation (2)).

$$T = \mu + 0.25 \times \sigma \times N \tag{2}$$

where $T$ denotes for thresholds, $\mu$ is the expected mean of dNDVI, $\sigma$ is the standard deviation of dNDVI, and $N$ ranges from 1 to 12.

Figure 4c shows the Z-scores, which are all positive. Pixels with higher Z-scores (red color) were not randomly distributed over the image area, and obvious spatial aggregation can be found in many artificial surface areas with high Z-scores. Figure 4d shows the histogram of the Z-scores. Judging from the curve of the histogram, it is one-sided with all positive values, only denoting the change magnitude and a majority of pixels falls close to zero. The values of thresholds were simulated using 12 decrement numbers: 1, 2, 3, 4, 5, 6, 10, 20, 30, 40, 50, 100.

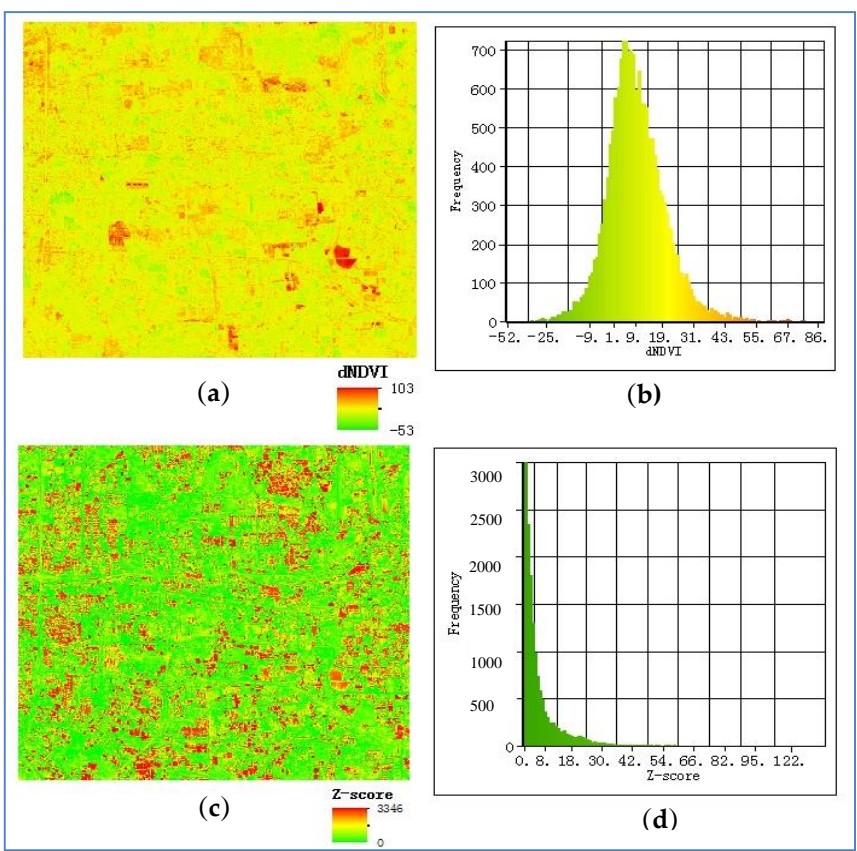

**Figure 4.** Spatial and frequency distributions of two change maps. (**a**) map of dNDVI; (**b**) histogram of dNDVI; (**c**) map of Z-score; (**d**) histogram of Z-score.

Simulations of thresholds on two different change interpreters were carried out on two different sampling designs based on the error matrix. Five indices were calculated, namely, the overall accuracy, omission error of C and NC, commission error of C as well as NC. Judging from the evaluation results in the random sampling (Figure 5a,b), we can find that for both thee two change variables, the trends in accuracy assessments were consistent: (1) with the increase of the thresholds, the overall accuracy increased dramatically. After certain thresholds, the overall accuracy reached the highest level, which was the proportion of the NC samples in the whole samples; (2) from the perspective of the producers, the C omission error decreased significantly, while the NC omission error increased dramatically as the threshold went higher; (3) from the perspective of users, the C commission error remained quite higher than the NC type; (5) the trend of overall accuracy had a corresponding tendency with the C

omission error. Looking at the number of C/NC samples (Figure 5c,d), it is clear that the higher the threshold values were, the fewer the C candidates and the more NC ones.

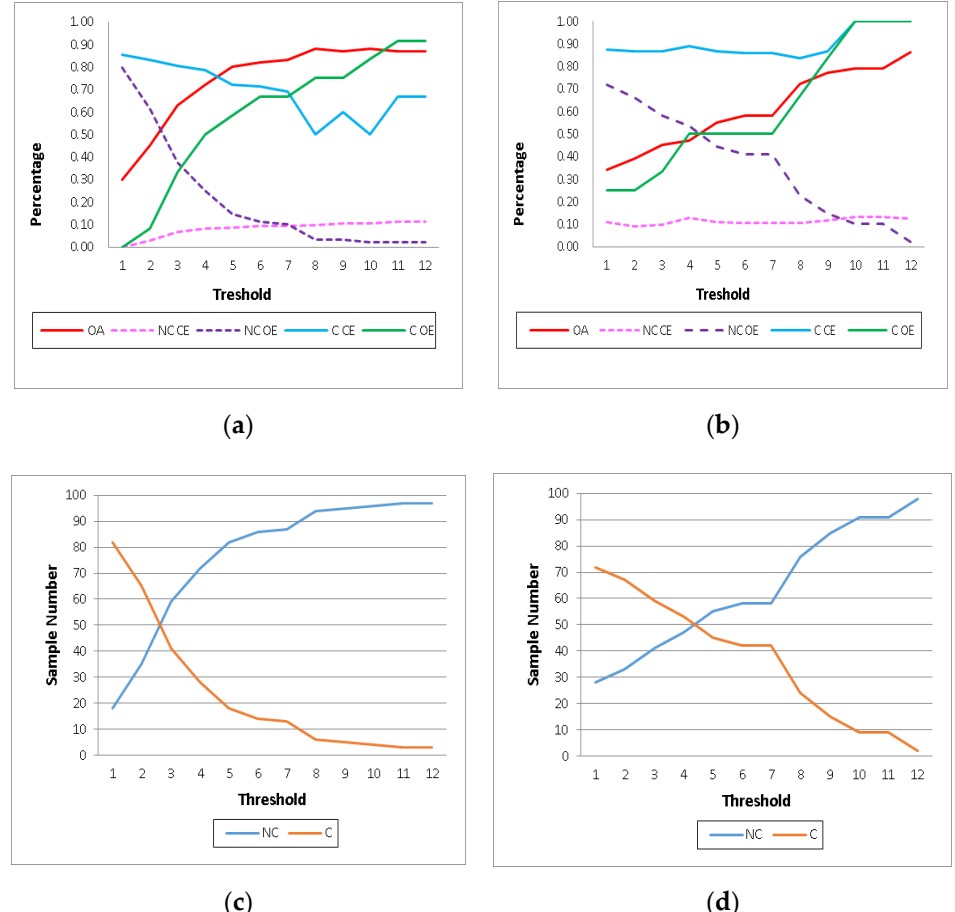

**Figure 5.** Trends in sensitivity analysis based on random sampling. (**a**) Accuracy assessments with the thresholds on dNDVI; (**b**) accuracy assessments with the thresholds on Z-score; (**c**) trends of C/NC samples (from users' side) with the thresholds on dNDVI; (**d**) trends of C/NC samples (from users' side) with the thresholds on Z-score. OA: Overall accuracy; C OE: omission error of change; C CE: commission error of change; NC OE: omission error of no change; NC CE: commission error of no change).

Looking at the accuracy assessments in stratified sampling (Figure 6a,b), for both the two change variables, the trends in accuracy assessments were also apparent: (1) with the increase of the thresholds, the overall accuracy kept a steady low level below 0.6; (2) similar to the previous experiments, the C omission error increased significantly, while the NC omission error decreased correspondingly; (3) unlike the pattern in the previous stage, the trend that the C commission error higher than the NC type was lost, instead they fluctuated at the same level; (4) the trend of overall accuracy had an excellent corresponding tendency with the NC commission error. Judging from the number of C/NC samples (Figure 6c,d), again, the higher the thresholds, the fewer C candidates and more NC ones. The sum of the numbers of the C/NC samples remained 100. In binary change detection, land change was put into the dichotic forms, so C and NC classes were complementary not only in image space but also in sample space.

Again, such significant trends in accuracy assessments can be viewed from two perspectives. From the users' perspective, which is the prediction side or the map side, when we increased the threshold on change magnitude, the predicted C candidates decreased, and the unqualified candidate will be the NC ones, as the pool of the samples remains the same. From the producers' perspective, the

C and NC samples were always the same. With the increase of the threshold, the readjusted predicted C and NC samples were further referenced to C/NC classes.

The above evaluation helps visualize a transparent accuracy assessment on ICD. To be mentioned, in Figures 5 and 6, a convergent point can be identified. The converged point for these four error (C OE/CE, NC OE/CE) lines seemed to denote a trade-off balance among different errors. For binary land change detection, the gain and loss of the C/NC sides had an effect on the accuracy assessment.

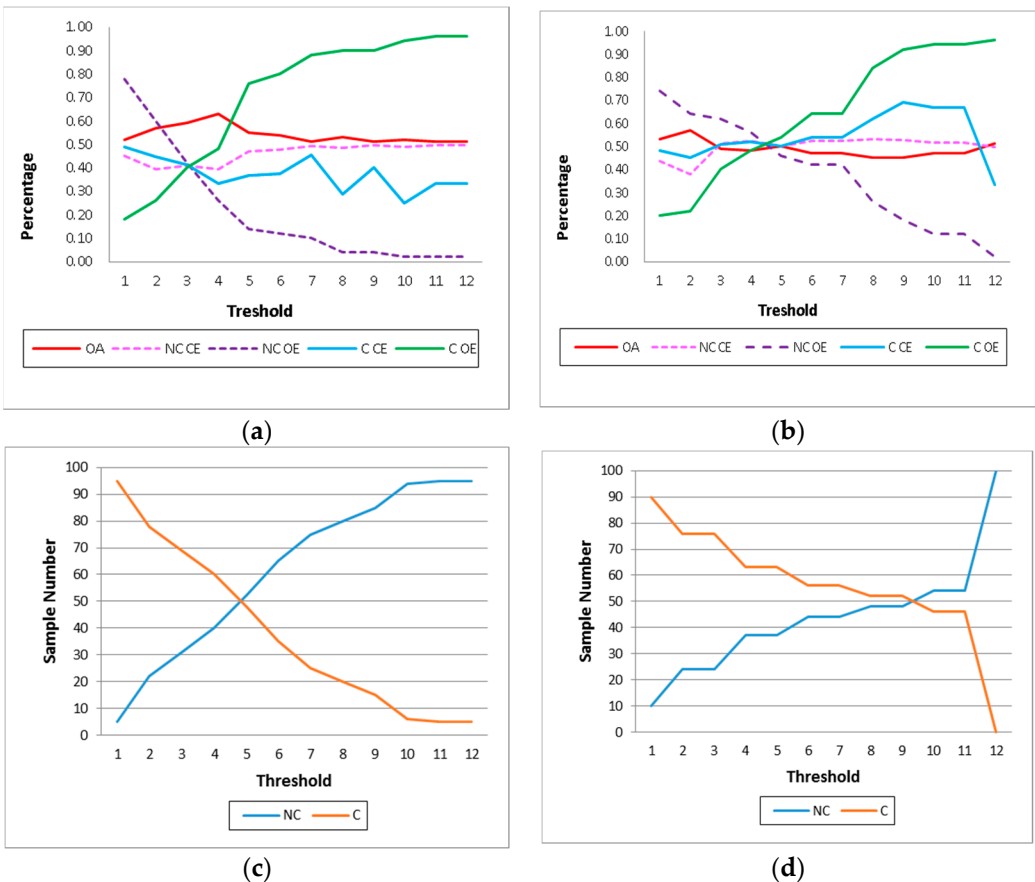

**Figure 6.** Trends in sensitivity analysis based on stratified sampling. (**a**) Accuracy assessments with the thresholds on dNDVI; (**b**) accuracy assessments with the threshold values established for the Z-score; (**c**) trends of C/NC samples (from users' side) with the threshold values established for the dNDVI; (**d**) trends of C/NC samples (from users' side) with the threshold values established for the Z-score.

## 4. Discussion

Errors and uncertainties are build-in imperfections in ICD and make it hard to make accurate decision in a chaotic situation (conflicting information from conflicting sources) [41]. As Mowrer [42] said, perhaps the worst nightmare of a natural resource manager is to appear "uncertain" to the public or to admit that there is an "error" in the decision process being presented. However, noise can give information on the state of the error source [43], and biases are as crucial as the correctness in land change detection. As too much emphasis had been put on the overall accuracy of a map, without considering the causes of the results [25], in this discussion, we shift our focus from the accuracy assessment to the causes of it by answering the following three questions: (1) How was the performance of the ICD methods? (2) Was the sampling biased or not? (3) Why are trends so significant in the accuracy analysis of binary land change detection?

### 4.1. Bi-Temporal Image Analysis Versus Image-and-Map Analysis

Two common binary change detection methods were adopted in this study, and their performance was expected to be judged from the accuracy analysis. If we simply judge from the overall accuracy, it seems that the performance of ICD under certain thresholds can be very good. Yet, after a careful check, reporting the overall accuracy with the omission and commission errors, it would show us that both of the two ICD methods in this study do not have a satisfying performance for land change detection. For bi-temporal image analysis, assumptions of identical image situations are critical. Yet, many assumptions were not carefully checked. For example, at the stage of image prepossessing, although photometric calibrations were adopted to minimize the effects of different sensors, Landsat solar geometry was not provided per pixel and TOA adjustment within a given scene did not vary per pixel [44], so TOA correction did not enhance within-scene fidelity. For image-and-map analysis, the classification map was regarded as the baseline in ICD so that the spectral center of land classes can be calculated. Yet, classification errors exist. Meanwhile, we assume that the majority of each class remains unchanged between paired images so that the spectral statistical features of land classes remain unchanged between two dates, which is also hard to guarantee.

When remote sensing images (time shots), which provide a snapshot of the land surface, are utilized to represent land dynamics, which are more appropriately viewed as fuzzy phenomena [23], uncertainty problems definitely exist. In binary change detection, land change dynamics is interpreted either as abrupt change (the complete replacement of one cover type by another, also called "real change") or as subtle change (the modification of land cover without changing its overall classification, also called "pseudo change") [5,31]. Subtle change has the spectral variance caused by the phenology problem (cyclical changes in the condition of the ground cover that do not relate to class change) [27]. Detection of image differences may be confused with problems in phenology and cropping, and such problems may be exacerbated by limited image availability, poor quality in temperate zones (especially clouds during critical times in spring of rapid phonological change), and difficulties in calibrating poor images [45]. Regions with high change magnitude are visualized in Figure 4a (vegetation area). It hints that the signal of phenology change highly interweaves with the signal of real land cover change. Without breaking these two types of signals apart, it is hard to present a satisfactory work. To minimize phenology effect, it is practical to adopt two pairs of images, one pair within leaf-on season and the other within leaf-off season to mask the area of subtle change [5].

### 4.2. Random Sampling Versus Stratified Sampling

Samples are regarded as a subset of individuals from within a population to estimate characteristics of the whole population [34]. To construct unbiased estimates, samples should concern the selection probabilities [46]. As land change is a rare feature of a given landscape, the C samples are scarce in the random sample pool. When thresholding is adopted, these samples have a greater chance of being detected as NC than C class. By setting high thresholds, the disproportion of C/NC samples can result in high overall accuracy, yet with high C omission and commission error and low NC omission and commission error. Although random sampling is a probabilistic sampling, the spatial and temporal heterogeneity of the study area is not considered. In this respect, random sampling cannot represent land change dynamics.

The optimal selection of spatial sampling locations depends in part on the spatial structure [47]. Many studies claimed that spatial stratified homogeneity can achieve more efficient spatial sampling and inference [34]. Theoretically, to achieve unbiased sampling, the evenness in the spatial-temporal domain should be considered. From the perspective of the spatial domain, the landscape is heterogeneous with various land classes; from the perspective of the temporal domain, the proportions of C/NC area are not equal. As our interest is in C/NC rather than from–to changes, two strata (i.e., C/NC) were chosen in stratified sampling. With the increase of the value of threshold, the trend of the trade-offs between C/NC errors still remains. Or in other words, the result is still biased. In this case, it is due to the abundant C samples. But stratified sampling has better performance than random sampling, as

smaller variance of overall accuracy has been identified. In this respect, stratified sampling is superior to random sampling.

### 4.3. Thresholding and Sensitivity Analysis

Thresholding is widely adopted in ICD, yet also widely criticized for its "hard/crisp" solutions [33]. First, it requires an arbitrary value of threshold, and there are few instances where this value can be rationally chosen. Secondly, our decisions are dichotomous. There are only two outcomes (i.e., C/NC). Because such techniques cannot quantify our degree of uncertainty, many researchers resorted to inferential statistical tools, such as hypothesis analysis, to achieve "soft" decisions [48–51]. In hypothesis analysis, a distribution for the observations is presented and then thresholds set on the point bearing certain a probability value showing the decision strength [51,52]. No matter what types of techniques are adopted for decision making, we want to be confident about our decisions. Unfortunately, it is impossible because the probability of two types of errors (i.e., Type I error and Type II error) cannot be simultaneously minimized [52]. Faced with this problem, statisticians adopt the strategy of controlling only $\alpha$ (level of significance), that is, to select a very low value for $\alpha$, the probability of making a Type I error [52] which is tantamount to saying "better to accept something false than reject something true". This is the same posture we had when we set the value of the thresholds. We started with a low value for the threshold, so that the probability of missing C pixels was low, as we were interested in change, and it was better for us not to set the value of the threshold high. To make a decision, costs are usually evaluated to compare the risk (change) of decisions [53]. As it is impossible for us to control C and NC errors simultaneously, not only in image space, but also sample space, it is a great challenge for us to determine the cost and risk.

Another inferential technique, sensitivity analysis (SA), a measure of the variations for a given input on a given output [54], has been emphasized as an essential part of modelling [55–57]. The advantage of such a technique is prominent because no assumptions are needed, compared to the technique of hypothesis analysis [58]. Sensitivity analysis allows modelers to improve and communicate the quality of their subjective beliefs about the merits of different strategies [59] and has been used in land surface change detection [4]. A practical way of SA on land change can be implemented by designing an unbiased sampling and calculating error matrix [30]. Similarly, in this study, the sensitivity of thresholds on land change decisions was evaluated, adopting the method of traditional accuracy assessments. Moreover, it is claimed that SA can identify the critical control point, because different options can be identified in such dynamic analysis, so that a decision maker can evaluate these options [59]. In this study, we set arbitrary values of thresholds to make land change decisions. As land change is rare, we intended to increase the values of the thresholds. If we have unbiased sampling, our experiments can help us to locate a critical point of setting the arbitrary value of the threshold in Figures 5 and 6, and it may be useful for decision making.

To be mentioned, both the careful sensitivity analysis of model-based inference and the paradox existing in its applications were emphasized in the literature [60]. In this study, significant trends were found in accuracy assessment. Yet, after further exploration on the mechanism of binary land change detection, we concluded that the failure to represent land change dynamics, especially for spatial sampling, would lead to dysfunctional accuracy analysis. To avoid perfunctory evaluation in binary change detection, it is necessary to find the causes of the results.

**Author Contributions:** Conceptualization and analysis, W.W.; funding acquisition, W.F.; writing—original draft, W.W. and M.H.-B.; writing—review & editing, W.W., C.W. and W.N.; supervision, M.H.-B., C.W. and W.F. All authors have read and agreed to the published version of the manuscript.

**Funding:** This work is mainly supported by the National Key Research and Development Program of China (No. 2018YFC1508803), and jointly supported by the National Key Research and Development Program of China (No. 2017YFA0604903). This work is also funded by Open Fund of State Laboratory of Information Engineering in Surveying, Mapping and Remote Sensing, Wuhan University (Grant No.19E01).

**Acknowledgments:** The authors would like to thank anonymous reviewers for their valuable comments and language editing which have greatly improved the quality of this manuscript.

**Conflicts of Interest:** No potential conflict of interest was reported by the authors.

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
