# Peer review of "Uncertainty Problems in Image Change Detection"

_sustainability, doi:10.3390/su12010274_

Round 1

Reviewer 1 Report

Thank you for the opportunity to read the manuscript "Perfunctory Sensitivity Analysis on Binary Land Change Detection". This is a paper that deals with an interesting issue and is well written regarding the content. I can recommend publication of this manuscript in "Sustainability" after a few minor issues are considered:

General Comment: I suggest the authors provide additional text in introduction, discussion and conclusion sections that relate their study with sustainability issues, which is the main topic of the journal.

Comment on the "Title": The title of the MS is ok.

Comment on the "Abstract" and "Keywords": Abstract and keywords are ok.

Comment on the "Introduction": The introduction section is well written, presenting the issue clearly. However, some minor issues are:

Line 33: Provide the reference in the appropriate for the journal style e.g.[1] I suggest the authors to clearly write at the end of Introduction the main aim and the specific objectives of their study. 

Comment on the "Materials and Methods": The proposed framework is thoroughly described. I recommend some minor changes:

A flow chart that describes the proposed framework will be useful.  I suggest the authors change the figures' 1 and 2 legends, as the letters are too small. 

Comment on the "Results": Results are well written.

Comment on the "Discussion": The discussion nicely explains and interprets the results. However, authors have to support their claims by adding additional literature in some parts e.g. Lines [252-254, 255-256, 256-261, 261-263, 264-267, 273-275, 287-290]

Comment on the Conclusion": Conclusions are ok.

Reviewer 2 Report

The manuscript presents a methodology to study changes in an area from a change detection procedure. The work is very simple and is based on the study of only two images, although it may be of interest to researchers. However, it presents a very succinct description of the results and they are not discussed in comparison with other similar existing procedures or other work on the same subject.

Some mistakes observed:

Line 33, reference bad formatted.

Line 42, abbreviature not defined previously.

Line 46, cite the literature, please.

Line 56, write space after times.

Line 56, explain better “Tremendous ICD methods…”

Line 94, write space after value.

Line 96, write space after robust.

In the introduction lacks a paragraph describing the objectives of the manuscript and the interest of the change detection technique. From 1995 there are many papers published about the subject, and it is necessary to point out the novelty that is presented.

The caption of figure 1 is poor describing the figure. The false RGB color can be explained here, not in figure; the small window has a text with very small font size. The position of the figure is not located in the boundaries of the margins of text. The box text claims about the coordinates system to be UTM, but the meridians are geographical coordinates; then latitude of origin has not sense.

The caption of figure 2 is broken in lines 129-130. Figure 2 box legend RGB is different from figure 1, please make the two RGB boxes with the same information about the bands.

Line 138, dNDVI is not defined before.

Line 142, explain what atmospheric correction have performed.

Line 149, write space after correlation.

Line 196, write space after matrix.

Line 204-217. The paragraph shows statements based on simple observations, without any scientific evidence. It identifies vegetation visually, without analysis of any kind. It finds no explanation for the distribution of water areas, and inadequately names the distribution in two tails. Finally, it states that the distribution of red pixels is not randomly distributed without performing a statistical distribution test, such as a Chi-square analysis.

Caption of figure 3 must be formatted correctly.

Figure 4. Caption bad formatted. Correct mistakes orthographic (Treshold), explain what is the number 1 in margin. The content of c and d is in a and b and can be merged.

Figure 5. Caption bad formatted. Correct mistakes orthographic (Treshold), explain what is the number 1 in margin.

The discussion needs a deep revision, because it lacks bibliography on which to discuss the presented results. Firstly, it should discuss its results with other similar matching results in similar works and secondly compare your work with other related work done in other places of the world. The reader expects at least 10 to 15 bibliographical references to prove that the results are consistent with others or different. The methodology proposed in the manuscript should be compared with other existing classification methodologies, implemented for remote sensing, and discuss the goodness if it exists between this work system and the other methodologies.

References must have the title of the journal abbreviated.

Reference 30 & 31 is the same.

References 34, 35 & 37 are bad formatted according to instructions for the journal.

Reviewer 3 Report

Typographical errors:

Line 15: change “on” to “of”

Line 18: change “special” to “spatial”

Lines 85-89: weak language and structure; vague sentence.

Line 90: do the authors mean “ICD” instead of “accuracy assessment”?

Line 117: is the selected image for 2010 a Landsat 5 or 7? Shouldn’t TM be changed to ETM throughout the manuscript?

Line 154-155: maintain consistency when using either Z or z for the z-score throughout the manuscript.

Line 181: what is the unit for the “sample size of 100”

Line 191: what is the unit for the “extent of 5x5”?

Line 207-208: “Part of the water area denotes great NDVI increment, which is difficult to explain” is a very confusing sentence and out of context.

Line 209: do you mean dNDVI in “The histogram of NDVI shows..”

Line 217: what is the unit for these numbers?

Lines 270-272: vague and convoluted (and may be contradicting) sentence.

Formatting errors:

Citation and numbering of References. I know the citation numbering was correct up to reference 14.. however, it seems one reference between number 15 and 21 was removed, and now all numbers should be reduced by 1.

All Figures, their descriptions should be listed after the Figure number and caption. In other words detailing (a), (b), ..etc. should be a part of the caption.

Literature citations:

Lines 59-60: provide a reference to the claim that “ICD algorithms are roughly grouped into two categories: image-based and image-map-based change analysis”.

Line 149-153: provide a reference for the details of Cross-Correlation algorithm.

Line 154: provide a reference for Equation 1

Line 174: provide a reference for equation 2.

Line 196-202: provide a reference for the details of accuracy measures.

Conceptual issues:

Line 65- 68: how is “image-map-based analysis” different from “post-classification analysis”? Why is the introduction of a new label?

Line 171: what is the motivation/reasoning for choosing 12 decrement?

Line 288: how can “the sensitivity analysis” help “in verifying and validating the ICD models great deal”? To my understanding, SA helps in assessing the impact of changes to the parameters on the outcome. In other words, SA assesses the sensitivity of the model to minor changes in the parameters; SA does not validate a model.

Overall assessment:

Through a combination of confused sentence structure, word choice, and verb tense, the authors failed to convince the reader of the importance and the practicality of the issue that they are dealing with and the contribution of their methodology in solving that issue.

Furthermore, using only one study area, the manuscript failed to indicate if the findings can be generalized to other situations and other parts of the world.

Sections 4 (Discussion) and 5 (Conclusion) are the weakest part of the manuscript as they seem to be mostly descriptive, trivial, and reflecting the authors opinions rather than linking their analysis to the outcome of the experiments.  

The concluding sentence on lines 290-292 states “In order to address such problem, it is necessary to report the overall accuracy, combining with the users’ accuracy (the commission error) and the producers’ accuracy (the omission error).” This is the most trivial and redundant sentence since this is the most widely used approach in practice.

Round 2

Reviewer 2 Report

It has been observed that before of [ there is not always a space, for example in lines 45, 46, 49, 54, 60, 61, 66, 67, 79, 90, 94 and many more. Correct where necessary.
In the conclusions, line 359, better explain in the text the phrase "Unfortunately, a perfunctory evaluation is highly suspected."
Bibliographic references should also be reviewed, as many of them are not formatted correctly. For example 2, 8, 11, 12, 12 and others.
The name of the journal is incorrect in some, such as number 1.
Check the format in number 19.
The name of the authors should be revised, as in 54.

Reviewer 3 Report

There is a good amount of improvement in the writing/presentation of this second version of the manuscript.

The addition of Figure 3 as the workflow of their methodology clarified some of the confusion in the methodology.

Typographical errors:

Line 17: change “national-wide” to “national” or “nation-wide”

Line 18: confusing “quality of these spatial infrastructures”; why is the switch from “products” to “infrastructure”?

Line 40: there is a confusion in the use of the terms LUCC (Land Use Cover Change) vs LC (Land Cover) or LULC (Land Use/Land Cover). References 2-4 talk about land cover databases, however, the authors (mistakenly or deliberately) presented this as “many regions have established LUCC database”. Same confusion is on Line 49.

Line 61: change “Candy” to “Canty”

Line 121: insert “for the study area” after “Thematic map and TM image”

Line 128: on the upper right block in Figure 3, do you mean “2010 Land cover Map” instead of “Land cover Classification”? Section 2.1 indicates that GlobaLand30 would be used; maybe the authors should stick to one choice of words for names and terminology.

Line 143: shouldn’t “2010 Landsat TM image” be changed to “GlobaLand30” or “2010 Land cover Map”?

Line 149: Change section number to 2.2.2 instead 2.2.3; unless there is a section is missing.

Line 156-157: unintelligible sentence “the higher the change magnitude is, the higher chance of having change will be”. Besides being grammatically wrong, it conveys no useful information. Do you mean that the higher the z-score? Or the higher the dNDVI? Or both?

Line 163: correct section number

Line 222: in Figure 5, the blue and green colors for CCE and COE lines are too close to separate between them, I suggest using a different pattern (e.g., dots) for one of them.

Line 264-267: very convoluted paragraph with, seemingly, some missing words.

Line 279: why is the switch from “image-map-based analysis” (line 64) to “map-and-image analysis”?

Line 283-284: confusing sentence, maybe switching the words “images” and “dates” in this sentence can improve it.

Line 360-361: unintelligible sentence, try simplifying the structure.

Line 362: “of” maybe missing in “from the perspective ICD modelling”

Line 418: incomplete list of authors for this reference.

Very unorthodox and uncommon use of the words that renders the sentence incomprehensible

Line 267: “hidden assumptions”

Line 301: “ICD is a highly developing area flood”

Line 313: “experience value”

Line 314: “experienced value”

Line 359: “perfunctory”

Line 363: “combing with”

Conceptual issues:

On lines 168-170, why is there a discussion on the actual land cover types in the study area if the interest is C/NC rather than from-to changes? If the interest is C/NC sampling, then how is “the simple random sampling” done?

Similarly, why do you need Google Earth (Line 174) if you are looking for C/NC samples?

Overall assessment:

The manuscript is not easy to read and understand. There is still a good amount of wrong word choices and verb tense, in addition to convoluted writing style.

The authors have not convinced the reader of the importance and the practicality of the issue that they are dealing with. There is no literature review whether other previous research was done on the issue at hand. As such, one can’t judge the novelty of their methodology nor evaluate its success.

With using only one study area, the manuscript failed to indicate if the findings can be generalized to other situations and other parts of the world.

Sections 4 (Discussion) is basically editorial, repetitive, and has no link to the findings and the analysis of the outcome of experiments. Except for a couple of sentences, the whole 100 lines in this section (line 263-356) can be completely removed with no impact on the manuscript. If the authors see values in their discussion, they can merge this section with section 1 (Introduction).
